# Path Integral Based Convolution and Pooling for Graph Neural Networks

**Zheng Ma**∗
Department of Physics
Princeton University
mazhengparnassum@gmail.com

**Junyu Xuan**∗
Centre for Artificial Intelligence
Faculty of Engineering and Information Technology
University of Technology Sydney
junyu.xuan@uts.edu.au

**Yu Guang Wang**∗
Max Planck Institute for Mathematics in the Sciences
& School of Mathematics and Statistics
University of New South Wales
yuguang.wang@mis.mpg.de

**Ming Li**
Department of Educational Technology
Zhejiang Normal University
mingli@zjnu.edu.cn

**Pietro Liò**
Department of Computer Science and Technology
University of Cambridge
Pietro.Lio@cl.cam.ac.uk

## Abstract

Graph neural networks (GNNs) extends the functionality of traditional neural networks to graph-structured data. Similar to CNNs, an optimized design of graph convolution and pooling is key to success. Borrowing ideas from physics, we propose a path integral based graph neural networks (PAN) for classification and regression tasks on graphs. Specifically, we consider a convolution operation that involves every path linking the message sender and receiver with learnable weights depending on the path length, which corresponds to the maximal entropy random walk. It generalizes the graph Laplacian to a new transition matrix we call *maximal entropy transition* (MET) matrix derived from a path integral formalism. Importantly, the diagonal entries of the MET matrix are directly related to the subgraph centrality, thus lead to a natural and adaptive pooling mechanism. PAN provides a versatile framework that can be tailored for different graph data with varying sizes and structures. We can view most existing GNN architectures as special cases of PAN. Experimental results show that PAN achieves state-of-the-art performance on various graph classification/regression tasks, including a new benchmark dataset from statistical mechanics we propose to boost applications of GNN in physical sciences.

## 1 Introduction

The triumph of convolutional neural networks (CNNs) has motivated researchers to develop similar architectures for graph-structured data. The task is challenging due to the absence of regular grids. One notable proposal is to define convolutions in the Fourier space [12, 11]. This method relies on finding the spectrum of the graph Laplacian $I - D^{-1}A$ or $I - D^{-\frac{1}{2}}AD^{-\frac{1}{2}}$ and then applies filters to

---

∗Equal contribution

the components of input signal $X$ under the corresponding basis, where $A$ is the adjacency matrix of the graph, and $D$ is the corresponding degree matrix. Due to the high computational complexity of diagonalizing the graph Laplacian, people have proposed many simplifications [17, 34].

The graph Laplacian based methods essentially rely on message passing [27] between directly connected nodes with equal weights shared among all edges, which is at heart a generic random walk (GRW) defined on graphs. It can be seen most obviously from the GCN model [34], where the normalized adjacency matrix is directly applied to the left-hand side of the input. In statistical physics, $D^{-1}A$ is known as the transition matrix of a particle doing a random walk on the graph, where the particle hops to all directly connected nodes with equiprobability. Many direct space-based methods [28, 40, 54, 63] can be viewed as generalizations of GRW, but with biased weights among the neighbors.

In this paper, we go beyond the GRW picture, where information necessarily dilutes when a path branches, and instead consider every path linking the message sender and receiver as the elemental unit in message passing. Inspired by the path integral formulation developed by Feynman [24, 23], we propose a graph convolution that assigns trainable weights to each path depending on its length. This formulation results in a *maximal entropy transition* (MET) matrix, which is the counterpart of graph Laplacian in GRW. By introducing a fictitious temperature, we can continuously tune our model from a fully localized one (MLP) to a spectrum based model. Importantly, the diagonal of the MET matrix is intimately related to the subgraph centrality, and thus provides a natural pooling method without extra computations. We call this complete path integral based graph neural network framework PAN.

We demonstrate that PAN outperforms many popular architectures on benchmark datasets. We also introduce a new dataset from statistical mechanics, which overcomes the lack of explanability and tunability of many previous ones. The dataset can serve as another benchmark, especially for boosting applications of GNN in physical sciences. This dataset again confirms that PAN has a faster convergence rate, higher prediction accuracy, and better stability compared to many counterparts.

## 2 Path Integral Based Graph Convolution

**Path integral and MET matrix**   Feynman's path integral formulation [24, 69] interprets the probability amplitude $\phi(x, t)$ as a weighted average in the configuration space, where the contribution from $\phi_0(x)$ is computed by summing over the influences (denoted by $e^{iS[\mathbf{x}, \dot{\mathbf{x}}]}$) from all paths connecting itself and $\phi(x, t)$. This formulation has been later extensively used in statistical mechanics and stochastic processes [35]. We note that this formulation essentially constructs a convolution by considering the contribution from all possible paths in the continuous space. Using this idea, but

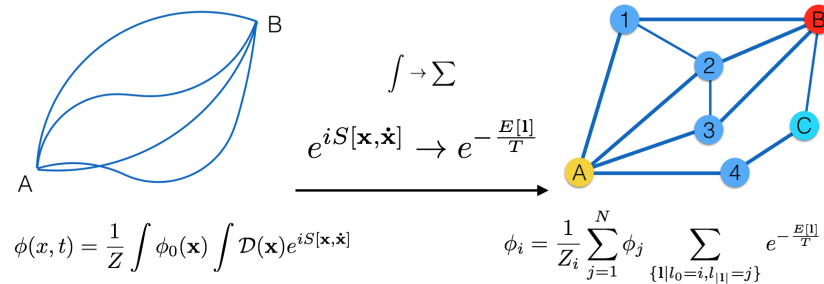

$$\phi(x,t) = \frac{1}{Z} \int \phi_0(\mathbf{x}) \int \mathcal{D}(\mathbf{x}) e^{iS[\mathbf{x}, \dot{\mathbf{x}}]} \qquad \phi_i = \frac{1}{Z_i} \sum_{j=1}^{N} \phi_j \sum_{\{\mathbf{l}|l_0=i, l_{|\mathbf{l}|}=j\}} e^{-\frac{E[\mathbf{l}]}{T}}$$

Figure 1: A schematic analogy between the original path integral formulation in continuous space (left) and the discrete version for a graph (right). Symbols are defined in the text.

modified for discrete graph structures, we can heuristically propose a statistical mechanics model on how information is shared between different nodes on a given graph. In the most general form, we write observable $\phi_i$ at the $i$-th node for a graph with $N$ nodes as

$$\phi_i = \frac{1}{Z_i} \sum_{j=1}^{N} \phi_j \sum_{\{\mathbf{l}|l_0=i, l_{|\mathbf{l}|}=j\}} e^{-\frac{E[\mathbf{l}]}{T}}, \qquad (1)$$

where $Z_i$ is the normalization factor known as the *partition function* for the $i$-th node. Here a path $\mathbf{l}$ is a sequence of connected nodes $(l_0 l_1 \dots l_{|\mathbf{l}|})$ where $A_{l_i l_{i+1}} = 1$, and the length of the path is denoted

by $|\mathbf{l}|$. In Figure 1 we draw the analogy between our discrete version and the original formulation. It is straightforward to see that the integral should now be replaced by a summation, and $\phi_0(x)$ only resides on nodes. Since a statistical mechanics perspective is more proper in our case, we directly change the exponential term, which is originally an integral of Lagrangian, to a Boltzmann's factor with fictitious energy $E[\mathbf{l}]$ and temperature $T$ (we choose Boltzmann's constant $k_B = 1$). Nevertheless, we still exploit the fact that the energy is a functional of path, which gives us a way to weight the influence of other nodes through a certain path. The fictitious temperature controls the excitation level of the system, which reflects that to what extent information is localized or extended. In practice, there is no need to learn the fictitious temperature or energy separately, instead the neural networks can directly learn the overall weights, as will be made clearer later.

To obtain an explicit form of our model, we now introduce some mild assumptions and simplifications. Intuitively, we know that information quality usually decays as the path between the message sender and the receiver becomes longer, thus it is reasonable to assume that the energy is not only a functional of path, but can be further simplified as a function that solely depends on the length of the path. Clearly, in principle one can incorporate information of individual edges, such as replacing the energy with a neural network that takes the nodes on a path as input. Therefore, the constant form used here should only be understood as the easiest implementation of our general framework. In the random walk picture, this simplification means that the hopping is equiprobable among all the paths that have the same length, which maximizes the Shannon entropy of the probability distribution of paths globally, and thus the random walk is given the name maximal entropy random walk [13]. [2] By first conditioning on the length of the path, we can introduce the overall $n$-th layer weight $k(n; i)$ for node $i$ by

$$k(n; i) = \frac{1}{Z_i} \sum_{j=1}^{N} g(i, j; n) e^{-\frac{E(n)}{T}}, \tag{2}$$

where $g(i, j; n)$ denotes the number of paths between nodes $i$ and $j$ with length of $n$, or *density of states* for the energy level $E(n)$ with respect to nodes $i$ and $j$, and the summation is taken over all nodes of the graph. Intuitively, node $j$ with larger $g(i, j; n)$ means that it has more channels to talk with node $i$, thus may impose a greater influence on node $i$ as the case in our formulation. For example, in Figure 1, nodes $B$ and $C$ are both two-step away from $A$, but $B$ has more paths connecting $A$ and would be assigned with a larger weight as a consequence. Presumably, the energy $E(n)$ is an increasing function of $n$, which leads to a decaying weight as $n$ increases.[3] By applying a cutoff of the maximal path length $L$, we exchange the summation order in (1) to obtain

$$\phi_i = \sum_{n=0}^{L} k(n; i) \sum_{j=1}^{N} \frac{g(i, j; n)}{\sum_{s=1}^{N} g(i, s; n)} \phi_j = \frac{1}{Z_i} \sum_{n=0}^{L} e^{-\frac{E(n)}{T}} \sum_{j=1}^{N} g(i, j; n) \phi_j, \tag{3}$$

where the partition function can be explicitly written as

$$Z_i = \sum_{n=0}^{L} e^{-\frac{E(n)}{T}} \sum_{j=1}^{N} g(i, j; n). \tag{4}$$

A nice property of this formalism is that we can easily compute $g(i, j; n)$ by raising the power of the adjacency matrix $A$ to $n$, which is a well-known property of the adjacency matrix from graph theory, i.e., $g(i, j; n) = A_{ij}^n$. Plug in (3) we now have a group of self-consistent equations governed by a transition matrix $M$ (a counterpart of the *propagator* in quantum mechanics), which can be written in the following compact form

$$M = Z^{-1} \sum_{n=0}^{L} e^{-\frac{E(n)}{T}} A^n, \tag{5}$$

where $\mathrm{diag}(Z)_i = Z_i$. We call the matrix $M$ *maximal entropy transition* (MET) matrix, with regard to the fact that it realizes maximal entropy under the microcanonical ensemble. This transition matrix replaces the role of the graph Laplacian under our framework.

More generally, one can constrain the paths under consideration to, for example, shortest paths or self-avoiding paths. Consequentially, $g(i, j; n)$ will take more complicated forms and the matrix $A^n$ needs to be modified accordingly. In this paper, we focus on the simplest scenario and apply no constraints for the simplicity of the discussion.

**PAN convolution**  The *eigenstates*, or the basis of the system $\{\psi_i\}$ satisfy $M\psi_i = \lambda_i \psi_i$. Similar to the basis formed by the graph Laplacian, one can define graph convolution based on the spectrum of MET matrix, which now has a distinct physical meaning. However, it is computationally impractical to diagonalize $M$ in every iteration as it is updated. To reduce the computational complexity, we apply the trick similar to GCN [34] by directly multiplying $M$ to the left hand side of the input and accompanying it by another weight matrix $W$ on the right-hand side. The convolutional layer is then reduced to a simple form

$$X^{(h+1)} = M^{(h)} X^{(h)} W^{(h)}, \tag{6}$$

where $h$ refers to the layer number. Applying $M$ to the input $X$ is essentially a weighted average among neighbors of a given node, which leads to the question that if the normalization consistent with the path integral formulation works best in a data-driven context. It has been consistently shown experimentally that a symmetric normalization usually gives better results [34, 41, 43]. This observation might have an intuitive explanation. Most generally, one can consider the normalization $Z^{-\theta_1} \cdot Z^{-\theta_2}$, where $\theta_1 + \theta_2 = 1$. There are two extreme situations. When $\theta_1 = 1$ and $\theta_2 = 0$, it is called random-walk normalization and the model can be understood as "receiver-controlled", in the sense that the node of interest performs an average among all the neighbors weighted by the number of channels that connect them. On the contrary, when $\theta_1 = 0$ and $\theta_2 = 1$, the model becomes "sender-controlled", since the weight is determined by the fraction of the flow coming out from the sender that is directed to the receiver. Because of the fact that for an undirected graph, the exact interaction between connected nodes are unknown, as a compromise, the symmetric normalization can outperform both extremes, even it may not be the optimal. This consideration leads us to a final perfection step that changes the normalization $Z^{-1}$ in $M$ to the symmetric normalized version. The convolutional layer then becomes

$$X^{(h+1)} = M^{(h)} X^{(h)} W^{(h)} = Z^{-1/2} \sum_{n=0}^{L} e^{-\frac{E(n)}{T}} A^n Z^{-1/2} X^{(h)} W^{(h)}. \tag{7}$$

We shall call this graph convolution *PANConv*.

The optimal cutoff $L$ of the series depends on the intrinsic properties of the graph, which is represented by temperature $T$. Incorporating more terms is analogous to having more particles excited to the higher energy level at a higher temperature. For instance, in *low-temperature limit*, $L = 0$, the model is reduced to the MLP model. In the *high-temperature limit*, all factors $\exp(-E(n)/T)$ are effectively one, and the term with the largest power dominates the summation. We can see it by noticing $A^n = \sum_{i=1}^{N} \lambda_i^n \psi_i \psi_i^T$, where $\lambda_1, \ldots, \lambda_N$ is sorted in a descending order. By the Perron-Frobenius theorem, we may only keep the leading order term with the unique largest eigenvalue $\lambda_1$ when $n \to \infty$. We then reach a prototype of the high temperature model $X^{(h+1)} = (I + \psi_1 \psi_1^T) X^{(h)} W^{(h)}$. The most suitable choice of the cutoff $L$ reflects the intrinsic dynamics of the graph.

## 3   Path Integral Based Graph Pooling

For graph classification and regression tasks, another critical component is the pooling mechanism, which enables us to deal with graph input with variable sizes and structures. Here we show that the PAN framework provides a natural ranking of node importance based on the MET matrix, which is intimately related to the subgraph centrality. This pooling scheme, denoted by PANPool, requires no further work aside from the convolution and can discover the underlying local motif adaptively.

**MET matrix and subgraph centrality**  Many different ways to rank the "importance" of nodes in a graph have been proposed in the complex networks community. The most straightforward one is the degree centrality (DC), which counts the number of neighbors, other more sophisticated measures include, for example, betweenness centrality (BC) and eigenvector centrality (EC) [45]. Although these methods do give specific measures of the global importance of the nodes, they usually fail to pick up local patterns. However, from the way CNNs work on image classifications, we know that it is the *locally* representative pixels that matter.

Estrada and Rodriguez-Velazquez [21] have shown that subgraph centrality is superior to the methods mentioned above in detecting local graph motifs, which are crucial to the analysis of many social and biological networks. The subgraph centrality computes a weighted sum of the number of self-loops with different lengths. Mathematically, it simply writes as $\sum_{k=0}^{\infty}(A^k)_{ii}/k!$ for node $i$. Interestingly, one immediately sees that the resemblance of this expression and the diagonal elements of the MET matrix. The difference is easy to explain. The summation in the MET matrix is truncated at maximal length $L$, and the weights for different path length $e^{\frac{E(n)}{T}}$ is learnable. In contrast, the predetermined weight $1/k!$ is a convenient choice to ensure the convergence of the summation and an analytical form of the result, which writes $\sum_{j=1}^{N} v_j^2(i)e^{\lambda_j}$, where $v_j(i)$ is the $i$-th element of the orthonormal basis associated with the eigenvalue $\lambda_j$.

Now it becomes clear that the MET matrix not only plays the role of a path integral-based convolution, its diagonal elements $M_{ii}$ also automatically provides a measure of the importance of node $i$, thus enabling a pooling mechanism by sorting $M_{ii}$. Importantly, this pooling method has three main merits compared to the subgraph centrality. First, we can exploit the readily-computed MET matrix, thus circumvent extra computations, especially the direct diagonalization of the adjacency matrix in the case of subgraph centrality. Second, the weights are data-driven rather than predetermined, which can effectively adapt to different inputs. Furthermore, the MET matrix is normalized [4], which adds weights on the *local* importance of the nodes, and can potentially avoid clustering around "hubs" that are commonly seen in real-world "scale-free" networks [8].

The PAN Pooling strategy has similar physical explanations as the PAN convolution. In the low-temperature limit, for example, if we set the cutoff at $L = 2$, the rank of $\sum_{n=0}^{L} e^{\frac{E(n)}{T}} A_{ii}^n$ is of the same order as the rank of degrees, and thus we recover the degree centrality. In the high-temperature limit, as $n \to \infty$, the sum is dominated by the magnitude of the $i$-th element of the orthonormal basis associated with the largest eigenvalue of $A$, thus the corresponding ranking is reduced to the ranking of the eigenvector centrality. By tuning $L$, PANPool provides a flexible strategy that can adapt to the "sweet spot" of the input.

To better understand the effect of the proposed method, in Figure 2, we visualize the top 20% nodes by different measures of node importance of a connected point pattern called RSA, which we detail in Section 5.2. It is noteworthy that while DC selects points relatively uniform, the result of EC is highly concentrated. This phenomenon is analogous to the contrast between the rather uniform diffusion in the classical picture and the Anderson localization [5] in the quantum mechanics of disordered systems [13]. In this sense, it tries to find a "mesoscopic" description that best fits the structure of input data. Importantly, we note that the unnormalized MET matrix tends to focus on the densely connected areas or hubs. In contrast, the normalized one tends to choose the *locally* representative nodes and leave out the equally well-connected nodes in the hubs. This observation leads us to propose an improved pooling strategy that balances the influencers at both the global and local levels.

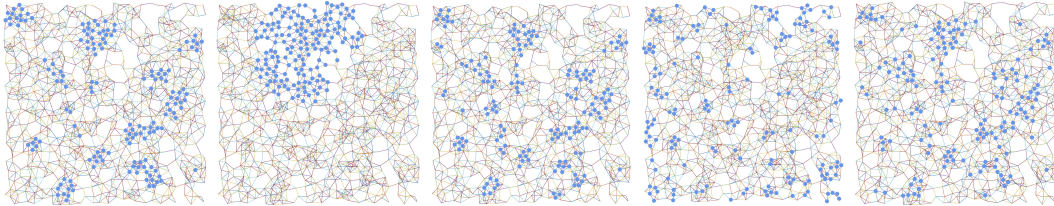

Figure 2: Top 20% nodes (shown in blue) by different measures of node importance of an RSA pattern from PointPattern dataset. From left to right are results from: Degree Centrality, Eigenvector Centrality, MET matrix without normalization, MET matrix and Hybrid PANPool.

**Hybrid PANPool** To combine the contribution of the local motifs and the global importance, we propose a hybrid PAN pooling (still referred as PANPool for simplicity) using a simple linear model. The global importance can be represented by, but not limited to the strength of the input signal $X$ itself. More precisely, we project feature $X \in R^{N \times d}$ by a trainable parameter vector $p \in R^d$ and

combine it with the diagonal $\mathrm{diag}(M)$ of the MET matrix to obtain a score vector

$$\text{score} = Xp + \beta\mathrm{diag}(M). \tag{8}$$

Here $\beta$ is a real learnable parameter that controls the emphasis on these two potentially competing factors. PANPool then selects a fraction of the nodes ranked by this score (number denoted by $K$), and outputs the pooled feature array $\widetilde{X} \in R^{K \times d}$ and the corresponding adjacency matrix $\widetilde{A} \in R^{K \times K}$. This new node score in (8) has jointly considered both node features (at global level) and graph structures (at local level). In Figure 2, PANPool tends to select nodes that are both important locally and globally. We also tested alternative designs under the same consideration, see supplementary material for details.

## 4 Related Works

Graph neural networks have received much attention recently [9, 40, 51, 60, 67, 68]. For graph convolutions, many works take accounts of the first order of the adjacency matrix in the spatial domain or graph Laplacian in the spectral domain. Bruna et al. [12] first proposed graph convolution using the Fourier method, which is, however, computationally expensive. Many different methods have been proposed to overcome this difficulty [6, 15, 16, 17, 27, 29, 34, 44, 52, 61, 62]. Another vital stream considers the attention mechanism [54], which infers the interaction between nodes without using a diffusion-like picture. Some other GNN models use multi-scale information and higher-order adjacency matrix [1, 2, 3, 25, 36, 41, 59]. Compared to the generic diffusion picture [28, 48, 53], the maximal entropy random walk has already shown excellent performance on link prediction [39] or community detection [47] tasks. However, many popular models can be related to or viewed as certain explicit realizations of our framework. We can interpret the MET matrix as an operator that acts on the graph input, which works as a kernel that allocates appropriate weights among the neighbors of a given node. This mechanism is similar to the attention mechanism [54], while we restrict the functional form of MET matrix based on physical intuitions and preserve a compact form. Although we keep the number of features unchanged by applying MET matrix, one can easily concatenate the aggregated information of neighbors like GraphSAGE [29] or GAT [54]. Importantly, the best choice of the cutoff $L$ reveals the intrinsic dynamics of the graph. In particular, by choosing $L = 1$, model (7) is essentially the GCN model [34]. The trick of adding self-loops is automatically realized in higher powers of $A$. By replacing $A$ in (7) with $D^{-1}A$ or $D^{-\frac{1}{2}}AD^{-\frac{1}{2}}$, we can easily transform our model to a multi-step GRW version, which is indeed the format of LanczosNet [41]. The preliminary ideas about PAN convolution have been presented at a previous ICML workshop [43], where the effectiveness of PAN convolution has been demonstrated by its superb performance on node classification tasks. Here we present a complete framework on both path integral based convolution and pooling mechanism, with a focus on classification and regression tasks at graph level.

Graph pooling is the other crucial component of GNNs to make the output uniform in size for graph classification and regression tasks. Researchers have proposed many pooling methods from different aspects. For example, one can merely consider node feature or node embeddings [20, 27, 55, 66]. These global pooling methods do not utilize the hierarchical structure of the graph. One way to reinforce learning ability is to build a data-dependent pooling layer with trainable operations or parameters [14, 26, 37, 38, 64]. One can incorporate more edge information in graph pooling [18, 65]. One can also use spectral method and pool in Fourier or wavelet domain [42, 46, 57]. PANPool is a method that takes both feature and structure into account. At last, it does not escape our analysis that the loss of paths could represent an efficient way to achieve dropout.

Finally, considering that PAN could learn the optimal representation of the intrinsic dynamics of the graphs by finding the optimal $L$, it is possible that PAN can alleviate the common information bottleneck of message-passing GNNs [4]. A relevant interesting theoretical problem could be comparing an extremely "wide" PAN against a deep GCN.

## 5 Experiments

In this section, we present the test results of PAN on various datasets in graph classification tasks. We show a performance comparison of PAN with some existing GNN methods. All the experiments

Table 1: Performance comparison for graph classification tasks (test accuracy in percentage for 10 repetitions; bold font is used to highlight the best performance in the list; the $L$ of all PAN-models on five datasets are $\{3, 1, 3, 3, 3\}$, respectively).

| Method | PROTEINS | PROTEINSF | NCI1 | AIDS | MUTAGEN |
|---|---|---|---|---|---|
| GCNConv + TopKPool | 64.0±0.40 | 69.6±6.03 | 49.9±0.50 | 81.2±1.00 | 63.5±6.69 |
| SAGEConv + SAGPool | 70.5±3.95 | 63.0±2.34 | **64.0±3.61** | 79.5±2.02 | 67.6±3.24 |
| GATConv + EdgePool | 72.4±1.46 | 71.3±3.16 | 60.1±1.76 | 80.5±0.72 | **71.5±1.09** |
| SGConv + TopKPooling | 73.6±1.70 | 65.9±1.25 | 61.5±5.11 | 81.0±0.01 | 66.3±2.08 |
| GATConv + ASAPooling | 64.8±5.43 | 67.3±4.37 | 53.9±4.11 | 84.7 ±6.21 | 58.4±5.19 |
| SGConv + EdgePooling | 69.0±1.74 | 70.5±2.48 | 58.4±1.96 | 76.7±1.12 | 70.7±0.69 |
| SAGEConv + ASAPooling | 59.2±5.84 | 63.9±2.44 | 53.5±2.91 | 80.6±6.39 | 63.1±3.74 |
| GCNConv + SAGPooling | 71.5±2.72 | 68.6±2.25 | 52.2±8.87 | 83.1±1.10 | 68.9±5.80 |
| PANConv+PANPool | **76.6±2.06** | **71.7±6.05** | 60.8± 3.45 | **97.5±1.86** | 70.9±2.76 |

were performed using PyTorch Geometric [22] and run on a server with Intel(R) Core(TM) i9-9820X CPU 3.30GHz, NVIDIA GeForce RTX 2080 Ti and NVIDIA TITAN V GV100. The codes can be downloaded at `https://github.com/YuGuangWang/PAN`.

## 5.1 PAN on Graph Classification Benchmarks

**Datasets and baseline methods**  We test the performance of PAN on five widely used benchmark datasets for graph classification tasks [33], including two protein graph datasets **PROTEINS** and **PROTEINS_full** [10, 19]; one mutagen dataset **MUTAGEN** [50, 32] (full name Mutagenicity); and one dataset that consists of chemical compounds screened for activity against non-small cell lung cancer and ovarian cancer cell lines **NCI1** [56]; one dataset that consists of molecular compounds for activity against HIV or not **AIDS** [50]. These datasets cover different domains, sample sizes, and graph structures, thus enable us to obtain a comprehensive understanding of PAN's performance in various scenarios. Specifically, the number of data samples ranges from 1,113 to 4,337, the average number of nodes is from 15.69 to 39.06, and the average number of edges is from 16.20 to 72.82, see a detailed statistical summary of the datasets in the supplementary material. We compare **PAN** in Table 1 with existing GNN models built by combining graph convolution layers **GCNConv** [34], **SAGEConv** [29], **GATConv** [54], or **SGConv** [58], and graph pooling layers **TopKPool**, **SAGPool** [38], **EdgePool** [42], or **ASAPool** [49].

**Setting**  In each experiment, we split 80%, 10%, and 10% of each dataset for training, validation, and testing. All GNN models share the same architecture: Conv($n_f$-512) + Pool + Conv(512-256) + Pool + Conv(256-128) + FC(128-$n_c$), where $n_f$ is the feature dimension and $n_c$ is the number of classes. We use batch size of 128 and 200 epochs. The learning rate and weightdecay are set to 0.01 and 5e-4, respectively. We evaluate the performance by the percentage of correctly predicted labels on test data. Specifically for PAN, we compared different choices of the cutoff $L$ (between 1 and 7) and reported the one that achieved the best result (shown in the description of Table 1).

**Results**  Table 1 reports classification test accuracy for several GNN models (see supplementary material for a larger table which includes results for aforementioned alternative pooling methods). PAN has excellent performance on all datasets and achieves top accuracy on three of the five datasets. In some cases, PAN improves state of the art by a few percentage points. For MUTAGEN, PAN still has the second-best performance. Most interestingly, the optimal choice of the highest order $L$ for the MET matrix varies for different types of graph data. It confirms that the flexibility of PAN enables it to learn and adapt to the most natural representation of the given graph data.

Additionally, we tested PAN on graph regression tasks such as QM7 and achieved excellent performances. See supplementary material for details.

## 5.2 PAN for Point Pattern Recognition

**A new benchmark dataset for graph classification**  People have proposed many graph neural network architectures; however, there are still insufficient well-accepted datasets to assess their relative strength [31]. Despite being popular, many datasets suffer from a lack of understanding of the

underlying mechanism, such as whether one can theoretically guarantee that a graph representation is proper. These datasets are usually not controllable either; many different prepossessing tricks might be needed, such as zero paddings. Consequentially, reproducibility might be compromised. In

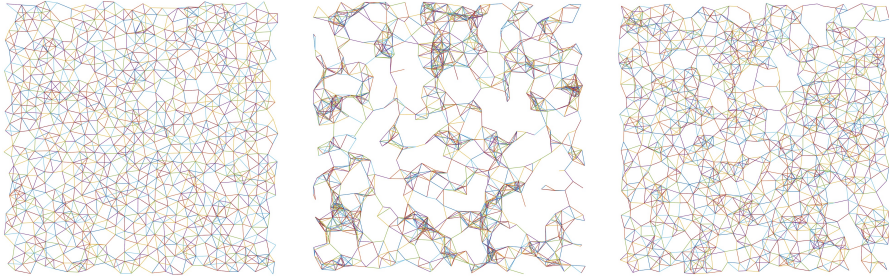

Figure 3: From left to right: Graph samples generated from HD, Poisson and RSA point processes in PointPattern dataset.

order to tackle this challenge, we introduce a new graph classification dataset constructed by simple point patterns from statistical mechanics. We simulated three point patterns in 2D: hard disks in equilibrium (HD), Poisson point process, and random sequential adsorption (RSA) of disks. The HD and Poisson distributions can be seen as simple models that describe the microstructures of liquids and gases [30], while the RSA is a nonequilibrium stochastic process that introduces new particles one by one subject to nonoverlapping conditions. These systems are well known to be structurally different, while being easy to simulate, thus provide a solid and controllable classification task. For each point pattern, the particles are treated as nodes, and edges are subsequently drawn according to whether two particles are within a threshold distance. We name the dataset **PointPattern**, which can be downloaded from the links contained in the supplementary material. See Figure 3 for examples of the three types of resulting graphs. The volume fraction (covered by particles) $\phi_{\mathrm{HD}}$ of HD is fixed at 0.5, while we tune $\phi_{\mathrm{RSA}}$ to control the similarity between RSA and the other two distributions (Poisson point pattern corresponds to $\phi_{\mathrm{RSA}}$=0). As $\phi_{\mathrm{RSA}}$ becomes closer to 0.5, RSA patterns are harder to be distinguished from HD patterns. We use the degree as the feature for each node. It thus allows us to generate a series of graph datasets with varying difficulties as classification tasks.

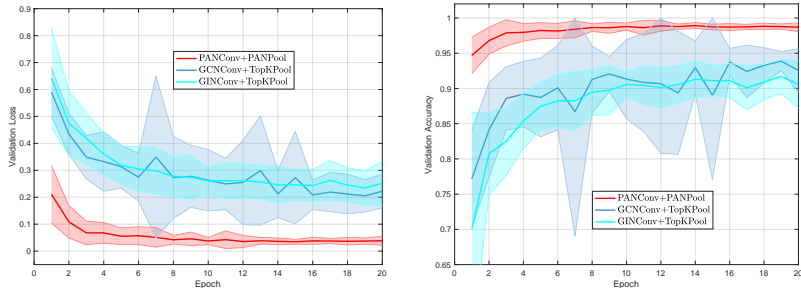

Figure 4: Comparison of validation loss and accuracy of PAN, GCN and GIN on PointPattern under similar network architectures with 10 repetitions.

**Setting** We tested the **PANConv+PANPool** model on **PointPattern** with $\phi_{\mathrm{RSA}} = 0.3, 0.35$ and $0.4$, and compared it with other two GNN models which use **GCNConv+TopKPool** or **GIN-Conv+TopKPool** as basic architecture blocks [14, 26, 34, 37, 62]. Each **PointPattern** dataset is a 3-classification problem for 15,000 graphs (5000 for each type) with sizes varying between 100 and 1000. All GNN models use the same network architecture: 3 units of one graph convolutional layer plus one graph pooling, followed by fully connected layers. In GCN and GIN models, we also use global max pooling to compress the node size to one before the fully connected layer. We split the data into training, validation, and test sets of size 12,000, 1,500, and 1,500. We fix the number of neurons in the convolutional layers to 64, the learning rate and weight decay are set to 0.001 and 0.0005.

**Results** Table 2 shows the mean and SD of the test accuracy of the three networks on the three PointPattern datasets. PAN outperforms GIN and GCN models on all datasets with 5 to 10 percents higher accuracy, while significantly reduces variances. We observe that PAN's advantage is persistent over varying task difficulties, which may be due to the consideration of higher order paths (here

Table 2: Test accuracy (in percentage) of PAN, GIN and GCN on three types of PointPattern datasets with different difficulties, epoch up to 20. The value in brackets is the cutoff of $L$.

| PointPattern | GINConv + SAGPool | GCNConv + TopKPool | PANConv + PANPool (ours) |
|---|---|---|---|
| $\phi_{\text{RSA}} = 0.3$ | 90.9±2.95 | 92.9±3.21 | 99.0±0.30 (4) |
| $\phi_{\text{RSA}} = 0.35$ | 86.7±3.30 | 89.3±3.31 | 97.6±0.53 (4) |
| $\phi_{\text{RSA}} = 0.4$ | 80.2±3.80 | 85.1±4.06 | 94.4±0.55 (4) |

$L = 4$). We compare the validation loss and accuracy trends in the training of PANConv+PANPool with GCNConv+TopKPool in Figure 4. It illustrates that the learning and generalization capabilities of PAN are better than those of the GCN and GIN models. The loss of PAN decays to much smaller values earlier while the accuracy reaches higher plateau more rapidly. Moreover, the loss and accuracy of PAN both have much smaller variances, which can be seen most evidently after epoch four. In this perspective, PAN provides a more efficient and stable learning model for the graph classification task. Another intriguing pattern we notice is that the weights are concentrated on the powers $A^3$ and $A^4$. It suggests that what differentiates these graph structures is the high orders of the adjacency matrix, or physically, the pair correlations at intermediate distances. It may explain why PAN performs better than GCN, which uses only $A$ in its model.

## 6  Conclusion

We propose a path integral based GNN framework (PAN), which consists of self-consistent convolution and pooling units, the later is closely related to the subgraph centrality. PAN can be seen as a class of generalization of GCN. PAN achieves excellent performances on various graph classification and regression tasks, while demonstrating fast convergence rate and great stability. We also introduce a new graph classification dataset **PointPattern** which can serve as a new benchmark.

## Broader Impact

The path integral based graph neural network presented in this paper provides a general framework for graph classification/regression tasks. We observe its advantages on the accuracy, convergence rate, and stability against many previous models. Given the physical ideas behind this framework, we believe PAN might be a powerful tool in analyzing biological, chemical, and physical systems. Specifically, the success over the simple point pattern dataset preludes its potentials in more sophisticated tasks such as detecting phase transitions and learning force fields in molecular dynamics, thus may accelerate materials discovery [7, 25]. Additionally, the study of PAN will potentially link the communities of both physics and machine learning. On the other hand, the PANPool strategy maintains a delicate balance on selecting representative nodes from both well-connected and "underrepresented" regions; this pooling method might be of particular interest to social scientists under specific contexts. However, one must be aware that the use of graph neural networks in commercial settings, such as recommendation systems, lending preferences, and fraud detection, may lead to negative ethical or social consequences. Since graph neural networks tend to relate a node's behavior to its environment, the abuse of this feature may, for example, enhance the growing opinion polarization in our society, and pose risks of systematic discrimination towards certain groups. Thus its use under these settings must be done with full mindfulness.

## Acknowledgments and Disclosure of Funding

The authors are grateful to the reviewers for their constructive comments. J. X. acknowledges the support from the Australian Research Council (ARC) under Grants No.: DE200100245. Y. W. acknowledges the support of funding from the European Research Council (ERC) under the European Union's Horizon 2020 research and innovation programme (grant agreement nº 757983). M. L. acknowledges support from the National Natural Science Foundation of China (No. 61802132 and 61877020).

## Footnotes

[2]For a weighted graph, a feasible choice for the functional form of the energy could be $E(l_{\mathrm{eff}})$, where the effective length of the path $l_{\mathrm{eff}}$ can be defined as a summation of the inverse of weights along the path, i.e. $l_{\mathrm{eff}} = \sum_{i=0}^{|l|-1} 1/w_{l_i l_{i+1}}$.

[3]This does not mean that $k(n; i)$ must necessarily be a decreasing function, as $g(i, j; n)$ grows exponentially in general. It would be valid to apply a cutoff as long as $E(n) \gg nT \ln \lambda_1$ for large $n$, where $\lambda_1$ is the largest eigenvalue of the adjacency matrix $A$.

[4]Notice that unlike the case in convolutions, the normalization being symmetric or not does not matter here. For pooling, we only care about the diagonal terms, and different normalization methods will give the same result.

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
