[Supplementary Material · NeurIPS_ID2329_SM.pdf]

# Path Integral Based Convolution and Pooling for Graph Neural Networks: Supplementary Material

**Zheng Ma**
Department of Physics
Princeton University
mazhengparnassum@gmail.com

**Junyu Xuan**
Centre for Artificial Intelligence
Faculty of Engineering and Information Technology
University of Technology Sydney
junyu.xuan@uts.edu.au

**Yu Guang Wang**
Max Planck Institute for Mathematics in the Sciences
& School of Mathematics and Statistics
University of New South Wales
yuguang.wang@mis.mpg.de

**Ming Li**
Department of Educational Technology
Zhejiang Normal University
mingli@zjnu.edu.cn

**Pietro Liò**
Department of Computer Science and Technology
University of Cambridge
Pietro.Lio@cl.cam.ac.uk

## 1  Variations of PANPool

In the main text, we discussed the relation between the diagonal of the MET matrix and subgraph centrality, as well as the idea of combining structural information and signals to develop pooling methods. We study several alternatives of the Hybrid PANPool proposed in the paper and report experimental results on benchmark datasets.

First, we consider the subgraph centrality's direct counterpart under the PAN framework, i.e., the weighted sum of powers of $A$. Formally, we consider the score as the diagonal of the MET matrix before normalization, it writes as

$$\text{score} = \text{diag}(Z^{\frac{1}{2}} M Z^{\frac{1}{2}}). \tag{1}$$

Similarly, we can also combine this unnormalized MET matrix with projected features, i.e.,

$$\text{score} = Xp + \beta \text{diag}(Z^{\frac{1}{2}} M Z^{\frac{1}{2}}). \tag{2}$$

This method also considers both graph structures and signals, while the measure of structural importance is at a global rather than local level.

We can also take simple approaches to mix structural information with signals. Most straightforwardly, we can employ the readily calculated convoluted feature $MX$ to define the score. For example, the $\ell_2$-norm of each row of $MX$ can define a score vector. The score for node $i$ can be written as

$$\text{score} = ||(MX)_i||_2. \tag{3}$$

Finally, instead of using a parameterized linear combination of the MET matrix and projected signals, we can apply the Hadamard product of the two contributions. The score then becomes

$$\text{score} = Xp \circ \text{diag}(M). \tag{4}$$

We use PANUMPool, PANXUMPool, PANMPool, and PANXHMPool to denote these variations of PANPool corresponding to (1)–(4) in the following experimental results.

## 2 Datasets and extended experiments

We put the PyTorch codes for experiments in the folder "codes" with dataset downloading and program execution instructions in "README.md".

### 2.1 PointPatterns

All simulations are performed in square simulation boxes with periodic boundary conditions. For hard disks, we use corresponding RSA configurations as initial conditions. We then perform an average of 10,000 Monte Carlo steps per particle to equilibrate the system. In the following step of converting a point pattern to a graph, we do not consider the images of the simulation boxes; that is, we do not connect particles across the boundaries. The choice of the threshold is inevitably subjective. Here we use $4R$ as the threshold, where $R$ is the radius of the corresponding hard disks with the same number density at volume fraction 0.5. This threshold is of the same order of the typical distance between two neighboring particles, which guarantees that the resulting graph is connected.

We list the summary statistics of the three datasets of PointPattern used in the main text with $\phi_{\mathrm{RSA}} = 0.3, 0.35, 0.4$ in Table 1. The datasets can be downloaded from Google Drive at

```
https://drive.google.com/file/d/1C3ciJsteqsKFVGF8JI8-KnXhe4zY41Ss/view?usp=
sharing
https://drive.google.com/file/d/1rsTh09FzGxHculBVrYyl5tOHD9mxqcOG/view?usp=
sharing
https://drive.google.com/file/d/16pI974P8WzanBUPrMHIaGfeSLoksviBk/view?usp=
sharing
```

We also show an example in README.md of running PAN on PointPattern, which includes downloading and preprocessing PointPattern datasets.

Table 1: Summary information of PointPattern datasets.

| PointPattern | $\phi_{\mathrm{RSA}} = 0.3$ | $\phi_{\mathrm{RSA}} = 0.35$ | $\phi_{\mathrm{RSA}} = 0.4$ |
|---|---|---|---|
| #classes | 3 | 3 | 3 |
| #graphs | 15,000 | 15,000 | 15,000 |
| max #nodes | 1000 | 1000 | 1000 |
| min #nodes | 100 | 100 | 100 |
| avg #nodes | 478 | 474 | 475 |
| avg #edges | 3265 | 3223 | 3220 |

### 2.2 PAN on Classification Benchmarks

**Extended experiments on Classification Benchmark** We list the summary statistics of benchmark graph classification datasets in Table 2. In Table 3, we report the classification test accuracy for variations of PAN compared with other methods. All networks utilize the same architecture. The PAN model, in general, has excellent performance on all datasets. The table shows the variations of PAN models can achieve the state of the art performance on a variety of graph classification tasks, and in some cases, improve state of the art by a few percentage points. In particular, PANConv+PANPool tends to perform better than other methods or variations on average, as presented in the main text. While among alternative PAN pooling methods, PANConv+PANMPool tends to have the least SD.

### 2.3 Quantum Chemistry Regression

**QM7** In this section, we test the performance of the PAN model on the QM7 dataset. The QM7 has been utilized to measure the efficacy of machine-learning methods for quantum chemistry [2, 9]. The dataset contains 7,165 molecules, each represented by the Coulomb (energy) matrix, and labeled with the value of atomization energy. The molecules have varying node size and structure with up to 23 atoms. We view each molecule as a weighted graph: atoms are nodes, and the Coulomb matrix of the molecule is the adjacency matrix. Since the node (atom) itself does not have feature information, we set the node feature to a constant vector with components all one, so that features

Table 2: Summary statistics of benchmark graph classification datasets.

| Dataset | MUTAG | PROTEINS | PROTEINSF | NCI1 | AIDS | MUTAGEN |
|---|---|---|---|---|---|---|
| max #nodes | 28 | 620 | 620 | 111 | 95 | 417 |
| min #nodes | 10 | 4 | 4 | 3 | 2 | 4 |
| avg #nodes | 17.93 | 39.06 | 39.06 | 29.87 | 15.69 | 30.32 |
| # node attributes | - | 1 | 29 | - | 4 | - |
| avg #edges | 19.79 | 72.82 | 72.82 | 32.30 | 16.20 | 30.77 |
| #graphs | 188 | 1,113 | 1,113 | 4,110 | 2,000 | 4,337 |
| #classes | 2 | 2 | 2 | 2 | 2 | 2 |

Table 3: Performance comparison for graph classification tasks (test accuracy in percentage; bold font is used to highlight the best performance in the list; the $L$ of all PAN-models on five datasets are $\{3, 1, 3, 3, 3\}$, respectively).

| Method | PROTEINS | PROTEINSF | NCI1 | AIDS | MUTAGEN |
|---|---|---|---|---|---|
| GCNConv + TopKPool | 64.0±0.40 | 69.6±6.03 | 49.9±0.50 | 81.2±1.00 | 63.5±6.69 |
| SAGEConv + SAGPool | 70.5±3.95 | 63.0±2.34 | **64.0±3.61** | 79.5±2.02 | 67.6±3.24 |
| GATConv + EdgePool | 72.4±1.46 | 71.3±3.16 | 60.1±1.76 | 80.5±0.72 | **71.5±1.09** |
| SGConv + TopKPooling | 73.6±1.70 | 65.9±1.25 | 61.5±5.11 | 81.0±0.01 | 66.3±2.08 |
| GATConv + ASAPooling | 64.8±5.43 | 67.3±4.37 | 53.9±4.11 | 84.7 ±6.21 | 58.4±5.19 |
| SGConv + EdgePooling | 69.0±1.74 | 70.5±2.48 | 58.4±1.96 | 76.7±1.12 | 70.7±0.69 |
| SAGEConv + ASAPooling | 59.2±5.84 | 63.9±2.44 | 53.5±2.91 | 80.6±6.39 | 63.1±3.74 |
| GCNConv + SAGPooling | 71.5±2.72 | 68.6±2.25 | 52.2±8.87 | 83.1±1.10 | 68.9±5.80 |
| PANConv+PANUMPool (Eq 1) | 67.8±0.82 | 69.1±1.21 | 59.2±0.69 | 82.7±7.82 | 70.0±2.11 |
| PANConv+PANXUMPool (Eq 2) | 69.7±1.60 | **72.6±3.20** | 60.1±1.74 | 86.9±3.64 | 69.4±1.08 |
| PANConv+PANMPool (Eq 3) | 66.8±0.78 | 71.0±0.60 | 51.9±1.39 | 80.6±0.44 | 68.4±1.01 |
| PANConv+PANXHMPool (Eq 4) | 68.8±5.23 | 69.7±1.97 | 55.9±1.81 | 91.4±3.39 | 70.2±1.08 |
| PANConv+PANPool | **76.6±2.06** | 71.7±6.05 | 60.8± 3.45 | **97.5±1.86** | 70.9±2.76 |

here are uninformative, and the learning is mainly concerned with identifying the molecule structure. The task is to predict the atomization energy value of each molecule graph, which boils down to a standard graph regression problem.

Table 4: Test mean absolute error (MAE) comparison on QM7, with the standard deviation over ten repetitions of the experiments. The value in brackets is the cutoff $L$.

| Method | Test MAE |
|---|---|
| Multitask [8] | 123.7±15.6* |
| RF [3] | 122.7±4.2* |
| KRR [4] | 110.3±4.7* |
| GC [1] | 77.9±2.1* |
| GCNConv+TopKPool | 43.6±0.98 |
| PANConv+PANUMPool (Eq 1) | 43.5±0.86 (1) |
| PANConv+PANXUMPool (Eq 2) | 43.3±1.32 (2) |
| PANConv+PANMPool (Eq 3) | 43.6±0.84 (2) |
| PANConv+PANXHMPool (Eq 4) | 43.0±1.27 (1) |
| **PANConv+PANPool** | **42.8±0.63** (1) |

'*' indicates records retrieved from [10], and bold font is used to highlight the best performance in the list.

**Experimental setting** In the experiment, we normalize the label value by subtracting the mean and scaling SD to 1. We then need to convert the predicted output to the original label domain (by re-scaling and adding the mean back). Following [5], we use mean squared error (MSE) as the loss for training and mean absolute error (MAE) as the evaluation metric for validation and test. We

use the splitting percentages of 80%, 10%, and 10% for training, validation, and testing. We set the hidden dimension of the PANConv and GCN layers as 64, the learning rate 5.0e-4 for Adam optimization, and the maximal epoch 50 with no early stop. For better comparison, we repeat all experiments ten times with randomly shuffled datasets of different random seeds.

**Comparison methods and results**    We test and compare the performance (test MAE and validation loss) of PAN against the GNN model with **GCNConv+SAGPool** [6, 7] and other methods including Multitask Networks (**Multitask**) [8], Random Forest (**RF**) [3], Kernel Ridge Regression (**KRR**) [4], Graph Convolutional models (**GC**) [1]. In our test, each PAN model contains one PANConv layer plus one PAN pooling layer, followed by two fully connected layers. The GCN model has two units of GCNConv+SAGPool, followed by GCNConv plus global max pooling and one fully connected layer. For other methods, we use their public results from [10] on QM7. In Table 4, we evaluate five PAN models on QM7 compared to other methods. The PAN models achieve top average test MAE and a smaller SD than other methods.