[Reviews · NeurIPS 2020]

Review 1

Summary and Contributions: ===AFTER AUTHOR FEEDBACK=== The responses successfully address my concern. I believe this work is novel and can make a valuable contribution to the literature. Hence I am willing to increase my overall score to 7. This paper proposes a new “convolution” method PANConv and a pooling method PANPool. Instead of considering each neighbor as current GNNs, PANConv view every path linking the message sender and receiver as the elemental unit in message passing process. This strategy can somehow avoid diluting information when aggregate information from distant node. It is shown that PANConv+PANPool can outperform current methods on graph-level tasks.

Strengths: (1) Intuitively, it is reasonable that considering every path as unit in message passing can probably alleviate the “Bottleneck” problem [1]. The motivation of PANConv is clear and meaningful. (2) The proposal of PANPool is convincing and the corresponding visualization (Figure 2) is great. (3) The experiments can demonstrate the effectiveness of PAN on graph-level tasks. [1] Alon, U., & Yahav, E. (2020). On the Bottleneck of Graph Neural Networks and its Practical Implications. arXiv preprint arXiv:2006.05205.

Weaknesses: (1) Although the motivation is claimed in this paper, it is not clear how this method can learn better representations than existing message passing methods. More intuitive descriptions, analysis examples, and/or theoretical supports should be included to show the superiority of the proposed method. (2) It would be more convincing if experiments on node classification tasks are utilized to show the effectiveness of the proposed convolution method. As we know, node classification is the most straight-forward usage of learned node representations. (3) More necessary ablation studies about PANConv and PANPool should be conducted, or we do not really know which part contributes to the improvement.

Correctness: The claims and method are correct and well described.

Clarity: In general, this paper is well written and clear. As mentioned before, it would be better if more intuitive descriptions, analysis examples, and/or theoretical supports should be included to show the superiority of the proposed method.

Relation to Prior Work: Yes. This paper describes the related work very consistently and the difference between previous work is well demonstrated.

Reproducibility: Yes

Additional Feedback: As shown in “Weaknesses”, more analysis and claim about why the proposed convolution can perform better than existing message passing methods is necessary for improvement. It would be more convincing if experiments on node classification and more ablation studies can be included.


Review 2

Summary and Contributions: This paper proposes a general framework for graph classification and regression tasks. The core idea, the path integral, which is borrowed from the quantum physics, is insightful and solid in theory. More importantly, this work links the communities of both physics and graph neural networks. Furthermore, this paper introduces a new dataset, which can serve as another benchmark.

Strengths: The proposed models in this paper, PAN convolution and PANPool is solid in theory. Given the physical ideas behind, PAN might be a powerful tool in analyzing many graph data, like biological, chemical, and physical systems. The authors also prove that many existing GNN methods (e.g. GraphSAGE, GAT, and GCN) are the special case of PAN. PAN is more accurate and stable compared with previous methods, which is demonstrated by the extensive experiments (Table 1 and Figure 4). Besides, the new dataset proposed in this paper can serve as another benchmark.

Weaknesses: The idea of PAN has been presented at the ICML workshop [1]. But that is not a big problem cause [1] focuses node-level classification, but this paper focuses graph-level classification and regression tasks instead and proposes a new model PANPool. Besides, the theoretical analysis is much more detailed. [1]Ma Z , Li M , Wang Y . PAN: Path Integral Based Convolution for Deep Graph Neural Networks[J]. 2019.

Correctness: The core idea is insightful. The experimental results also demonstrate the claims.

Clarity: The organization of the paper is good. The paper is well written and the language is rigorous and standardized.

Relation to Prior Work: This work clearly discusses the differences from the previous work in the section of related work. Moreover, the authors prove that many existing GNN models can be viewed as special cases of PAN convolution (line 206~211).

Reproducibility: Yes

Additional Feedback: Post rebuttal After reading the authors rebuttal, I think this is a qualified paper for NeurIPS and keep my rating unchanged.


Review 3

Summary and Contributions: This paper proposed a novel mechanism of graph convolution and graph pooling based on a discrete analogy of the path integral formulation used in Physics. The resultant convolution operator leverages different powers of adjacency matrices in a simple way. Authors further propose a graph pooling method based on the normalization constant in the path-integral formulation which captures the subgraph centrality to some extent. Experiments on several graph classifications tasks and a newly proposed point distribution recognition task (along with a new dataset) show that the proposed method achieves very impressive results. ---------- I read the response from authors and other review comments. My concerns are largely resolved. I still believe this is a solid contribution to the community and would like to keep my original rating.

Strengths: 1, I like the connection between the path integral and graph convolution. The proposed framework indeed provides a simple yet elegant way of leveraging multi-scale information in the network/graph. Moreover, this framework permits a simple graph pooling formulation. From my understanding, the proposed method is easy to implement and just adds little overheads to the vanilla GCN. 2, The related work is adequately discussed. 3, The paper is clearly written. I enjoy reading the paper. The intuitive examples about the normalization strategy and the pooling indeed help the understanding.

Weaknesses: 1, One “concern” or suggestion I had is about the expressiveness of the proposed PANConv operator. Besides the regular weights, the only learnable “weight” for the n-th power of the adjacency matrix is a scalar, i.e., e^{-E(n)/T}. From a pure machine learning perspective, I think it would be better to have more capacity on the weight by using a vector or even matrix parameterization (the entries are output by some neural networks which not only takes path length n as input but also some feature from A^n) of the weight although I am not sure about its physical interpretation. 2, It would be great to run some baselines without using the proposed pooling so that one can understand how important the pooling is. 3, Although Hybrid PAN is proposed, I did not find any experimental verification on its usefulness in the paper. 4, In section 5.1, there is no validation set mentioned for the graph classification task. It is a bit surprising that L = 1 or 2 already performs so well. It is unclear from the description whether PANConv of all layers share the same cutoff L. If that is the case, the same cutoff L at top layer may have a different effect than the same L at bottom layer since bottom layers already contribute to the diffusion of the node information. Therefore, I feel like the comparison of L not only depends on the graph type but also the architecture (e.g., depth). 5, In section 5.2, since one of the contributions is this dataset, I would suggest authors to introduce some of the important aspects of it like the statistics. By just looking at the figure 3, it seems that the number of nodes under each category is quite different and may be exploited by the model easily. Later, I checked the supplementary file and found the average number of nodes under three categories are almost the same which relieved my worry. But it would be great to spend some space in the paper to showcase the statistics like table 1 in the supplementary file. 6, The newly proposed PointPattern dataset seems very easy which may not be a good benchmark. Do you have any thoughts to increase the level of difficulty? 7, Figure 3 is a bit small.

Correctness: From my understanding, the caims and the method are correct.

Clarity: The paper is clearly written.

Relation to Prior Work: It is clearly discussed regarding the differences from previous work.

Reproducibility: Yes

Additional Feedback:


Review 4

Summary and Contributions: ===AFTER AUTHOR FEEDBACK=== The author feedback addresses my comments on hyperparameters and node classification. I still stand by my original review otherwise, and believe this is a solid paper that belongs in NeurIPS This paper proposes a graph convolution method, PAN inspired by path integrals. In a PAN convolution step, each node is influenced by all paths between that node and its neighbors, weighted by the path length. When viewed as a random walk of information along the graph, this operation corresponds to a walk that maximizes entropy rather than a generic random walk, as is the case with GCN. The weight matrix associated with PAN can be used to downsample the graph based on subgraph centrality as well. Experiments show that PAN outperforms several other common graph convolution methods graph classification and regression tasks where graphs represent molecules. They also show PAN's superior performance on a new artificial dataset they created.

Strengths: The algorithm presented in this paper is novel and well motivated with meaningful connections to existing graph theory/network concepts. Learning from graph data with deep networks is currently a very relevant and active topic in the Neurips community. The empirical results are significant and support the the argument that path number and length are useful quantities with which to weight neighbor node contributions.

Weaknesses: This method as presented and implemented still has the limited power and expressiveness of message passing GNNs which take weighted averages over neighboring nodes.

Correctness: The claims and method appear to be correct. The only potential issue that I noticed, is that there is no validation set for the graph classification benchmarks, and it's unclear how the hyperparameters were chosen for the different networks.

Clarity: The paper is clear and well written.

Relation to Prior Work: Yes, Section 4 contextualizes this paper among existing graph convolution work.

Reproducibility: No

Additional Feedback: The paper states that model hyperparameters for the graph classification task are provided in the supplementary material. I do not see them there. Please add them. All of the experiments with the exception of the proposed synthetic dataset are graph classification/regression on molecule data, but PAN should apply just as naturally to network data and node classification. Why did you choose only moceules? If PAN also performs very well at node classification on network graph data, that would be impressive. I'm also curious about using PAN with edge features, our more complex learned energy functions. Have you experimented with anything along this line?

[Author Response · NeurIPS 2020]

We thank all the reviewers for acknowledging the novelty and concreteness of the path-integral idea presented in this paper. We find it very encouraging that the reviewers found the presentation clearly connects our graph learning framework with its deep roots in physics, and that the experimental results are convincing. We indeed believe in the conceptual and practical values of the PAN framework. We hope the new data set can also contribute to the community. We are very grateful for the reviewers' thoughtful and constructive comments. By incorporating many of these suggestions, we believe the quality of the manuscript is further improved. We discuss the major comments of the reviews, as follows.

**PANConv for node classification (Reviewers #1, #2, #4)**  We appreciate Reviewers #1 and #4's interests in the performance of PANConv on node classification tasks, which we also agree are the best places to test the effectiveness of the convolution unit. In fact, as Reviewer #2 pointed out (also mentioned in related works), the advantage of the PAN convolution idea in node classification tasks have been reported in a short ICML workshop paper, see arXiv:1904.10996. The authors showed that path-integral based convolution could achieve excellent/top performance on benchmark Citation Networks datasets as compared to many existing convolution methods such as GCN. Moreover, this convolution method converges faster, which is consistent with what we found here in graph classification tasks. The current work distinguishes itself from the workshop paper by focusing on graph-level tasks and providing a pooling method. Additionally, as Reviewer #2 kindly pointed out, the discussion here is much more in-depth and more detailed. A new data set from statistical physics is also presented.

**Ablation study (Reviewers #1, #3)**  We thank Reviewer #1 for this great suggestion. As we mentioned above, the effectiveness of PANConv is already observed in node classification tasks. We have run extra experiments and here we show some preliminary results to this end. In Table 1 (shown below), we fix the convolution unit (PANConv)/pooling unit (PANPool) and replace the other unit with TopKPool/GCNConv, respectively. We find with PANConv or PANPool, the GNNs achieve superior performance, which provides further evidence on the effectiveness of both units. We hope these results also help to answer Reviewer #3's question on the usefulness of PANPool alone.

Table 1: Ablation test for PANConv (L=4) and PANPool on graph classification benchmarks

| Method | PROTEINS | PROTEINSF | NCI1 | AIDS | MUTAGEN |
|---|---|---|---|---|---|
| **PANConv**+TopKPool | 61.61 | 59.82 | 55.47 | 93.50 | 57.70 |
| GCNConv+**PANPool** | 66.96 | 66.07 | 66.67 | 88.50 | 66.90 |
| **PANConv**+**PANPool** | 69.64 | 70.24 | 74.30 | 99.50 | 68.82 |

**More complex energy/weight functions (Reviewers #3, #4)**  This is a fascinating question which we would like to test in future. Considering the scope of the paper, for simplicity, here we only assign a scalar to the weight of paths with a certain length. This practice should be understood as the easiest implementation of the general PAN framework. We do not see fundamental difficulties in extending the weights to incorporate information of edges, such as replacing them with a neural network that takes the nodes on a path as input.

**Some clarifications on the new dataset, codes, etc (Reviewer #3, #4)**  For Reviewer #3, in the generation process of point patterns, the number of nodes for each class is drawn from the same distribution. Figure 3 simply shows some samples with similar numbers of nodes (736, 703, 784, respectively). We will add more details in the supplementary material (SM) as well as in the paper if space permits. As mentioned in Section 5.2, when we increase the value of $\phi$ closer to 0.5, it is harder to distinguish the RSA pattern from HD. It thus adds to the difficulty of the classification. We may also add more complex point processes in the future (e.g., random organization), but their interpretations are less straightforward. In the main text, we have used "Hybrid PANPool" in all experiments. So "PANPool" is used as a general term but also refers to Hybrid PANPool in implementations. In the SM, we also studied many variations of PANPool. The cutoff $L$ for each layer is chosen to be the same. In principle, there is no constraint on it, but this practice significantly simplifies the training.

For Reviewer #4, we provided the PyTorch codes for demonstration in the SM. More details are contained in README.md, which shows how to run the program. We have prepared a Github page and will release it immediately after the review process. We also updated the validation information in the main text.

**Ongoing/Future work: Information Bottleneck (Reviewers #1, #4)**  We are very thankful to Reviewer #1 for directing us to the interesting reference on information bottleneck, which we now cite. This is a common challenge for all GNNs, which echoes Reviewer #4's comments on the limitation of expressiveness of all message-passing GNNs. Interestingly, Reviewer #1 pointed out that PAN may actually alleviate this problem compared to other methods. Guided by this insight, we are now carrying out experiments for different depths. We will include those results if time permits. We also plan to study the comparison between a "wide" PAN and a deep GCN in the future.

[Meta-Review · NeurIPS 2020]

All reviewers unanimously support accepting this paper and the work seems really solid. I hope the authors can take the reviewer suggestions into account when revising the paper for the final version, for example adding the new results in the rebuttal to the paper to make it better.